# Minimum Description Length Recurrent Neural Networks

## Abstract

Recurrent neural networks (RNNs) face two well-known challenges: (a) the difficulty of such networks to generalize appropriately as opposed to memorizing, especially from very short input sequences (generalization); and (b) the difficulty for us to understand the knowledge that the network has attained (transparency). We explore the implications to these challenges of employing a general search through neural architectures using a genetic algorithm with Minimum Description Length (MDL) as an objective function. We find that MDL leads the networks to reach adequate levels of generalization from very small corpora, improving over backpropagation-based alternatives. We demonstrate this approach by evolving networks which perform tasks of increasing complexity with absolute correctness. The resulting networks are small, easily interpretable, and unlike classical RNNs, are provably appropriate for sequences of arbitrary length even when trained on very limited corpora. One case study is addition, for which our system grows a network with just four cells, reaching 100% accuracy (and at least .999 certainty) for arbitrary large numbers.

## 1 Introduction

The modeling of sequential knowledge and learning requires making appropriate generalizations from input sequences that are often quite short. This holds both for language capabilities and for other sequential tasks such as counting. Moreover, it is often helpful for the modeler to inspect the acquired knowledge and reason about its properties. Neural networks, despite their impressive results and popularity in a wide range of domains, still face some challenges in these respects: they tend to overfit the learning data and require regularization or other special measures, as well as very large training corpora, to avoid this problem. In terms of knowledge, networks are often very big, and it is generally very hard to inspect a given network and determine what it is that it actually knows (see Papernot & McDaniel, 2018, among others, for a recent attempt to probe this knowledge).

Some of the challenges above arise from the reliance of common connectionist approaches on backpropagation as a training method, and in this paper we explore the implications to sequential modeling of well-known alternative perspectives on neural network design. Specifically, we consider replacing backpropagation with a general search using a genetic algorithm through a large space of possible networks using Minimum Description Length (MDL; Rissanen, 1978) as an objective function. In essence, this amounts to minimizing error as usual, while at the same time trying to minimize the size of the network. We find that MDL helps the networks reach adequate levels of generalization from very small corpora, avoiding overfitting and performing significantly better than backpropagation-based alternatives. The MDL search converges on networks that are often small, transparent, and provably correct. We illustrate this across a range of sequential tasks.

## 2 Previous work

Our work follows several lines of work in the literature. Evolutionary programming has been used to evolve neural networks in a range of studies. Early work that uses genetic algorithms for various aspects of neural network optimization includes Miller et al. (1989), Montana & Davis (1989), Whitley et al. (1990), and Zhang & Mühlenbein (1993; 1995). These works focus on feed-forward architectures, but Angeline et al. (1994) present an evolutionary algorithm that discovers recurrent

neural networks and test it on a range of sequential tasks that are very relevant to the goals of the current paper. Evolutionary programming for neural networks remains an active area of research (see Schmidhuber, 2015 and Gaier & Ha, 2019, among others, for relevant references).

In terms of objective function, Zhang & Mühlenbein (1993; 1995) use a simplicity metric that is essentially the same as the MDL metric that we use (and describe below). Schmidhuber (1997) presents an algorithm for discovering networks that optimize a simplicity metric that is closely related to MDL. Simplicity criteria have been used in a range of works on neural networks, including recent contributions (e.g., Ahmadizar et al., 2015 and Gaier & Ha, 2019).

Our paper connects also with the literature on using recurrent neural networks for grammar induction and on the interpretation of such networks in terms of symbolic knowledge (often formal-language theoretic objects). These challenges were already taken up by early work on recurrent neural networks (see Giles et al., 1990 and Elman, 1990, among others), and they remain the focus of recent work (see, e.g., Wang et al., 2018 and Weiss et al., 2018). See Jacobsson (2005) and Wang et al. (2018) for discussion and further references.

In a continuation of these efforts, our contribution is twofold. First, we put together a minimal, out-of-the-box combination of the core of these ideas: evaluate the performance that can be achieved by a learner that seeks to optimize MDL measures. The search and optimization itself is done through a standard genetic algorithm. From there, we benchmark the performance obtained through MDL optimization against the performance obtained by a set of 3 classic RNN architectures of different sizes. Second, we show the benefit of optimizing networks not only for performance but also for their own architecture size, in that it makes the black box much more permeable; for current tasks, we are able to provide full proofs of accuracy (above and beyond a test set).

## 3 LEARNER

### 3.1 MDL

Consider a hypothesis space $\mathcal{G}$ of possible grammars, and a corpus of input data $D$. In our case, $\mathcal{G}$ is the set of all possible network architectures expressible using our representations, and $D$ is a set of input sequences. For a given $G \in \mathcal{G}$ we may consider the ways in which we can encode the data $D$ given that $G$. The MDL principle (Rissanen, 1978), a computable approximation of Kolmogorov Complexity (Solomonoff, 1964; Kolmogorov, 1965; Chaitin, 1966), aims at the $G$ that minimizes $|G| + |D : G|$, where $|G|$ is the size of $G$ and $|D : G|$ is the length of the shortest encoding of $D$ given $G$ (with both components typically measured in bits). Minimizing $|G|$ favors small, general grammars that often fit the data poorly. Minimizing $|D : G|$ favors large, overly specific grammars that overfit the data. By minimizing the sum, MDL aims at an intermediate level of generalization: reasonably small grammars that fit the data reasonably well. MDL – and the closely related Bayesian approach to induction – have been used in a wide range of models of linguistic phenomena, in which one is often required to generalize from very limited data (see Horning, 1969, Berwick, 1982, Stolcke, 1994, Grünwald, 1996, and de Marcken, 1996, among others).

The term $|D : G|$ corresponds to the surprisal of the data $D$ according to the probability distribution defined by $G$ and is closely related to the cross-entropy between the distribution defined by $G$ and the true distribution that generated $D$. The term $|G|$ depends on an encoding scheme for a network. We provide the details of such an encoding scheme in Appendix A and now turn to describe the space of networks that will be considered.

### 3.2 REPRESENTATIONS

A network is represented as a directed graph which contains nodes, weighted edges, and activation functions for each node. Since we do not use backpropagation to train the network, the set of possible networks is larger here than what is usually allowed; for example, output units can have outgoing edges which feed hidden units, and input units can feed into other input units, thus saving intermediate hidden units. Beyond the topological flexibility, the activation functions also allow for more diversity in the possible networks, they can vary freely from one unit to the next, and they can be chosen from any set of possible activation functions, including non-differentiable ones

since training does not rely on backpropagation. Currently, we only allow four possible activation functions: identity (i.e., no activation function), square function, ReLU, and sigmoid.

Since we are ultimately interested in sequential tasks, we add a second type of edges – recurrent edges – which cross time steps and feed a unit with the value of another unit at the previous step. Such edges are required in order to create memory cells and counters for various sequential tasks.[1]

### 3.3 SEARCH

Given our use of MDL as an objective function, which is not differentiable, and our aim of optimizing the network structure itself rather than just the weights of a fixed architecture, gradient-based training methods such as backpropagation would not naturally support this objective.

Instead, we use a Genetic Algorithm (GA; Holland, 1975) which frees us from the constraints coming from backpropagation and fulfills the two requirements at once. For simplicity and to highlight the utility of the MDL metric as a standalone objective, we use a vanilla implementation of GA. The GA advances by incrementally evolving networks (e.g. add an edge, adjust a weight, remove a unit, etc.), and ranks them by their MDL score. Full implementation details are given in Appendix B.[2]

### 3.4 INPUT AND OUTPUT

In all tasks, the learner is fed with inputs from a sequence, one input after the next, and at each time step its outputs are interpreted as determining the probability to obtain a particular output. Depending on the task, this output is deterministically or probabilistically derivable from the inputs up to this point, and it may or may not correspond to the next input in the sequence.

If the vocabulary contains $n$ letters, the inputs are one-hot encoded over $n$ input cells (in yellow in the figures), and the outputs are given in $n$ cells (in blue). To interpret these $n$ outputs as a probability distribution we zero negative values and normalize the rest to sum to 1. In case of a degenerate network that outputs all 0's, the probabilities are set to the uniform value $1/n$. When the vocabulary is binary, we use a single 0/1-valued input cell and a single output cell whose value is interpreted as the probability to obtain 1, clipping it to the $[0, 1]$ range if necessary.

### 3.5 ILLUSTRATION WITH ELEMENTARY, DETERMINISTIC TASKS

First, let us provide a simple illustration, the **identity** task: the network is fed with a random sequence of binary digits, and the target output is identical to the input at each time step. The network developed by the MDL learner is given in Fig. 1(a); this network is transparent and shows that this very simple task is learned perfectly well. Anticipating on a number of baselines, more results are given visually in Appendix E and numerically in Appendix F, where it can be seen that though some (but not all) classic RNNs may perform this task well, they do not achieve a perfect performance, only a statistically good one (that is, assigning a high probability to the appropriate value, but not necessarily assigning it a probability of 1).

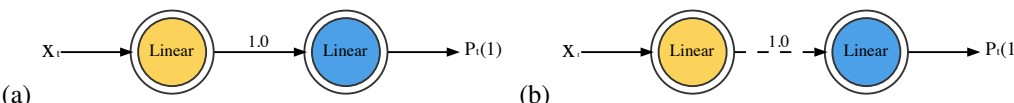

(a)  (b)

Figure 1: The networks found by the MDL learner for (a) the **identity** task and (b) the **previous character task** (same result for all training sets of length 10, 20, 50, and 100). The input (arriving to the yellow cell on the left) is directly fed into the output (blue cell on the right) with a weight of 1, either through a direct, 'contemporary' connection (a), or through a cross-time connection in (b).

Second, consider the **previous character** task: the network is fed with a random sequence of binary digits, and the output is identical to the *previous* input at each time step. This requires the learner

---

[1]Another specificity of this representation is that cells may have no input at all (which is potentially beneficial in terms of the grammar length part of the MDL score). In such cases, the cell behaves as if it received a total input of 0. So for instance if the activation function is a sigmoid, the output is constantly 0.5.

[2]The model source code and experimental material will be published once the paper can be de-anonymized.

to develop some kind of memory. The network developed by the MDL learner is given in Fig. 1(b). Again, one can comprehend how this network produces its output, and the classic RNNs tested do not perform as well (and aren't as transparent).

## 4    ARTIFICIAL LANGUAGE-MODELING EXPERIMENTS

### 4.1    SETUP

A convenient choice to test an MDL learner and its generalization capabilities comes from language induction tasks, in which the corpus is generated from a formally well-identified generalization. We ran tasks based on several classical formal-language learning challenges from the linguistic domains of syntax and semantics. Let us mention two aspects in which these tasks differ from deterministic tasks as the ones above. First, they concern the prediction of the **next character in a sequence** (not about predicting an independent output based on the input). Second, the next character — that is, the target output — was **not a deterministic function** of the input (or all inputs so far), but followed in the training and test set from a more general probability distribution. Given the results above, this could be seen as a challenge for the MDL learner which seems to make categorical decisions.

### 4.2    BASELINES

To compete with our MDL learner, we trained 12 standard RNNs, varying in their architecture (**GRU** cells, **LSTM** cells, **Elman** cells) and the size of their hidden state vector (2, 4, 32, 128). As usual, a final softmax, derivable layer was plugged at the end of these networks for them to output a well-formed probability distribution. These RNNs were trained with a cross-entropy loss.[3]

Additionally, we added two abstract baselines: a **uniform** baseline corresponding to predicting all outputs with equal probability in all cases, and an **optimal** baseline, corresponding to predicting the actual probability distribution, that is the one determined by how the task was set up.

### 4.3    EXPERIMENTS

In each of the tasks below the training corpora consist of several sequences of the form $s_1 \# s_2 \# \ldots$, where the $s_i$'s are strings over an alphabet that does not include $\#$. As before, the task is to read each character in a sequence and to predict the next character. The structure of the $s_i$'s corresponds to various formal language-theoretical regularities, as described below. In sections 4.3.1 to 4.3.3 these regularities come from the domain of the semantics of quantificational determiners (Barwise & Cooper, 1981; see Tiede, 1999 and Paperno, 2011, among others, for a discussion of the learnability of such patterns). The formal languages in these tasks are regular and can be dealt with by finite-state automata. In sections 4.3.4 and 4.3.5 we consider patterns of unbounded counting based on a classic syntactic challenge (Gers & Schmidhuber, 2001) and correspond to context-free and context-sensitive languages, which require more expressive frameworks. All results are given visually in Appendix E and numerically in Appendix F.

### 4.3.1    EXACTLY $n$

In this task, each $s_i$ is made of zero or more 0's, and of exactly $n$ 1's: at each time step, the next input is 0 or 1 with equal probability if the current $s_i$ has fewer than $n$ 1's and it is 0 or $\#$ if there are exactly $n$ 1's already. The order of the 0's and 1's is thus random. For 'exactly 3', for example, one possible $s_i$ would be 001011000. The model trained on sequences of length 100, 200, 500, and 1,000, and tested on an unseen sequence of length 1,000.

The MDL learner achieves a test cross-entropy of 1580, 1293, 997.2, 997.2 for the four training sets, the lowest possible cross-entropy being 997. Against the RNN alternatives, the MDL networks are ranked 2/13, 2/13, 1/13, 1/13. Fig. 2 shows the network found for $n = 1$ and the largest training set of length 1,000.

---

[3]All RNNs were trained using the Adam optimizer (Kingma & Ba, 2015) with learning rate 0.01, $\beta_1 = 0.9$, and $\beta_2 = 0.999$. The networks were trained by feeding the full batch of training data for 1,000 epochs.

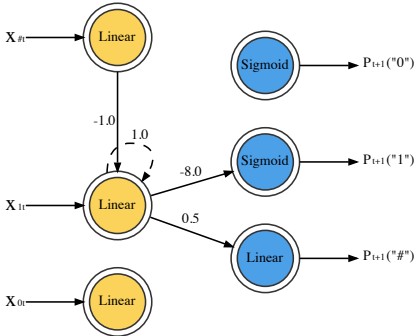

Figure 2: The network found by the MDL learner for the **exactly 1** task. The network keeps track of the number of 1's seen so far in the middle input cell (see the recurrent connection that makes it persists in memory), and resets it when a # is input (see the -1 connection from the top input cell).

### 4.3.2 AT LEAST $n$

In this task, each $s_i$ is made of zero or more 0's, and of $n$ or more 1's: at each time step, the next input is 0 or 1 with equal probability if the current $s_i$ has fewer than $n$ 1's, and it is 0, 1 or # if there are exactly $n$ 1's already, also with equal probability. The order of the 0's and 1's is thus random. For 'at least 3', for example, one possible $s_i$ would be 01010110010.

The model trained on sequences of length 200, 500, and 1,000, and tested on a single unseen sequence of length 1,000. For $n = 1$, the MDL learner achieves a test cross-entropy of 1582, 1458, 1346 for the three training sets (lowest possible cross-entropy is 1345). Against the RNN alternatives, the networks are ranked 2/13, 3/13, 1/13. Fig. 3(a) shows the network found for $n = 1$ and the largest training set.

### 4.3.3 BETWEEN $m$ AND $n$

Here each $s_i$ has zero or more 0's, and between $m$ and $n$ 1's: at each time step, the next input is 0 or 1 with equal probability if the current $s_i$ has less than $m$ 1's, it is 0, 1 or # if there are between $m$ and $n - 1$ 1's already, also with equal probability, and it is 0 or 1 if the number of 1's has reached $n$. The order of the 0's and 1's is thus random. For 'between 3 and 6', for example, one possible $s_i$ would be 01010110010.

This was tested with $(m, n) = (3, 6)$. The model trained on sequences of length 100, 200, 500, and 1,000, and tested on an unseen sequence of length 1,000. The MDL learner achieves a cross-entropy of 1580, 1394, 1175, 1176 for the four training sets (lowest possible cross-entropy is 1159). Against the RNN alternatives, the MDL networks are ranked 1/13, 2/13, 1/13, 1/13. Fig. 3(b) shows the network found for the largest training set.

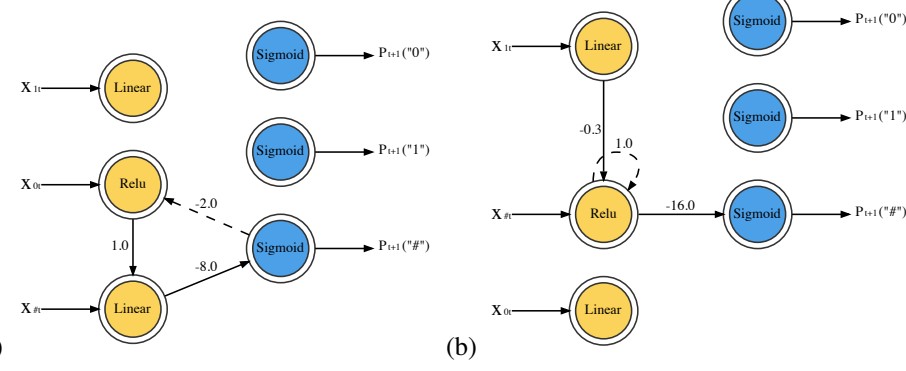

(a)          (b)

Figure 3: The network found by the MDL learner for (a) the **at least 1** task and (b) the **between 3 and 6** task for the largest training set.

### 4.3.4  $a^n b^n$

Here, each $s_i$ belongs to the context-free language $a^n b^n$, where $n \geq 0$. In order to recognize the language, an unbounded counter needs to be developed.

When generating the sequences, the next character is $a$ with probability .9; after a series of $a$'s the sequence switches to $b$'s with probability .1, and all remaining symbols are deterministically fixed by the fact that we aim for an $a^n b^n \#$ sequence.

The model trained on sets consisting of 10 or 100 sequences and tested on an unseen set of 1,000 sequences. The MDL learner achieves a test cross-entropy of 9829 and 4755 for the two training sets (lowest possible cross-entropy is 4680). Against the RNN alternatives, the networks are ranked 4/13 and 1/13. In Fig. 4(a) we show the network found for the larger training set.

### 4.3.5  $a^n b^n c^n$

Each $s_i$ belongs to the context-sensitive language $a^n b^n c^n$, where $n \geq 0$. In order to succeed, the network needs to keep in memory the number of $a$'s seen so that it can deterministically predict the moment to switch from $b$'s to $c$'s, and from $c$'s to the end of sequence symbol. The model was trained on sets consisting of 10 or 100 sequences, randomized similarly to the previous task. The final network was tested on an unseen set of 1,000 sequences. Fig. 4(b) shows the network found for the larger training set. In Appendix C, we show more precisely that the network assigns a probability of .91 or above to the correct output at any deterministic time step,[4] for *any* value of $n$. The MDL learner thus achieves a stable accuracy of 100% (on the test set and in fact for any relevant sequence), and a test cross-entropy of 4987 (when the lowest possible cross-entropy is 4680). Against the RNN alternatives, the network is ranked 3/13.

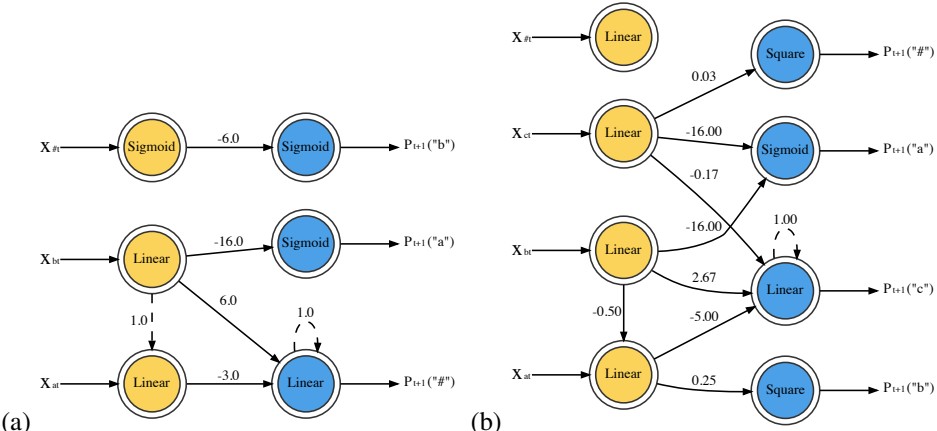

(a)  (b)

Figure 4: The network found by the MDL learner for (a) the $a^n b^n$, and (b) the $a^n b^n c^n$ task for the largest training set. In (a), the bottom right cell takes care of the counting: it decreases at each new input $a$ (by increments of 3), and decreases at each new increment of $b$ (by increment of 3: +6 directly from the middle input cell and -3 indirectly through the bottom left input cell). In network (b), the self-loop cell handles counting of all three symbols, first by decreasing by the number of $a$'s, then increasing for each $b$, then decreasing again for $c$'s; the output cell values at each time step align with the counter's value to create the correct probability distribution.

---

[4]The end of the initial sequence of $a$'s cannot be deterministically predicted.

## 4.4 OVERVIEW OF THE RESULTS

| Task | Optimal cross-entropy | MDL cross-entropy | Best RNN, no. hidden units | Best RNN cross-entropy | MDL rank vs. RNN |
|---|---|---|---|---|---|
| Exactly 1 | 997 | 997.2 | Elman, 2 | 1009 | 1/13 |
| At least 1 | 1345 | 1346 | Elman, 2 | 1366 | 1/13 |
| Between 3-6 | 1159 | 1176 | GRU, 2 | 1178 | 1/13 |
| $a^n b^n$ | 4680 | 4755 | LSTM, 4 | 4805 | 1/13 |
| $a^n b^n c^n$ | 4680 | 4987 | LSTM, 128 | 4830 | 3/13 |
| Addition | 0 | 173 | Elman, 4 | 9050 | 1/13 |

Table 1: Results overview: cross-entropy and ranking of the MDL learner compared to the RNN alternatives (here for the largest training set in each task, see details in Appendices E and F).

Table 1 provides an overview of the results, full details are given visually in Appendix E and numerically in Appendix F. As illustrated in the figures above, the networks that the MDL learner finds are sufficiently **small and transparent** that their workings can be inspected directly. In each case, this network expresses a pattern that is either identical to the one that was used to generate the corpus or is very close to it.

Not surprisingly, this translates to **good performance** in terms of cross-entropy. Even though the learner did not attempt to optimize cross-entropy directly, the cross-entropy of the MDL network is close to the entropy of the true distribution across several corpus sizes. Sometimes the two are almost the same, but even when this is not the case the MDL network performs no worse (and typically much better) than the random baseline.

Things are different with the comparison RNNs from the literature. These networks are large and opaque, and they perform unreliably: occasionally one of them performs well for a particular corpus size, but others will typically perform much worse than chance, and which architecture does what can change significantly for the next corpus size or the next task. Overall, the MDL learner performs best on the test set in 21 of the 40 tasks and training conditions presented here,[5] while the next best learner wins only 7 of the remaining tasks.

To the extent that we can identify a trend in the performance of the RNNs it is that the best performance generally comes from small networks, with few hidden units (for 26 tasks out of 40, the winner among the RNNs has 2 hidden units, the minimal number possible here). Smaller networks may perform worse than bigger ones on the training corpus, but they generalize better and perform better at test.[6] This is of course in line with the intuition behind our own learner and the MDL approach more broadly.

## 5 CASE STUDY: GENERAL ADDITION

Recent advancements in large-scale language models such as GPT-3 (Brown et al., 2020) have brought attention to the capability of such models to generalize as opposed to memorize. One particular test case is that of general addition, which humans tackle with relative ease using few examples, but that is not picked up in full generality by any deep learning learner to our knowledge.[7,8]

---

[5]Even in cases where the MDL learner is not the winner its cross-entropy is close to that of the winner or to the optimal baseline; in fact, sometimes the RNN winner has a cross-entropy below that of the optimal baseline, which makes it a suspicious winner.

[6]As a result, it means that picking the right RNN architecture is not an easy task (without a tripartite training/dev/test set), and performance at training is not a good predictor of performance at test.

[7]In other people's words: "As far as I know there is no neural network that is capable of doing basic arithmetic like addition and multiplication on a large number of digits based on training data rather than hard-coding.", Kevin Lacker, 'Giving GPT-3 a Turing Test'. https://lacker.io/ai/2020/07/06/giving-gpt-3-a-turing-test.html

[8]GPT-3, one of the most recent and ambitious models, succeeds at many tasks but not quite addition: "One particularly interesting case is arithmetic calculation: the model gives a perfect score for 2-digit addition and

In the setting we explore, the network is fed with sequences of pairs of binary digits, representing the digits of two binary numbers to be added up. The output at each step is the corresponding digit of the sum of these two numbers. With a small training set of all pairs of integers up to 10 (total 100 samples), the MDL learner fails because in some cases it predicts a categorical probability (a plain 0 or 1) for the wrong output. With a larger training set of 400 samples (all pairs up to 20), the MDL learner develops the network given in Fig. 5. The network achieves 100% accuracy on any test set, with cross-entropy 173 compared to an optimal 0.

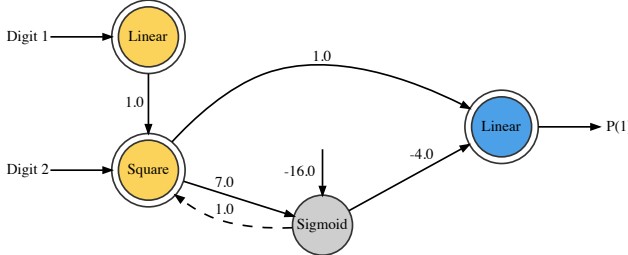

Figure 5: The network found by the MDL learner for the **addition** task, trained on 400 pairs of numbers. The first digit is added to the second digit, and the sum is squared in place. Next, a hidden cell (in yellow) with a recurrent connection was evolved to take care of the carry-over. The network reaches 100% accuracy on a test set consisting of all pairs of numbers up to 250, and is in fact provably correct for any arbitrary pair of numbers.

Here again, this network is quite transparent. In short, the output $h_n$ of the hidden cell (in yellow) at any given time step $n$ corresponds to the carry-over. (With $i_n$ and $j_n$ the inputs, $h_n = \mathrm{sigmoid}(7(i_n + j_n + h_{n-1})^2 - 16)$, and this goes to $1$ — that is, there is a carry-over — if the sum of the inputs and the carry-over from the previous time step is large enough). In Appendix D, we show more precisely that the network assigns a probability of .999 or above to the correct output under all circumstances.

Again, the task is learned very well and in a readable fashion. None of the comparison RNNs that we consider do as well, coherent with observations made above. To our knowledge no other RNN has been proven to hold a carry-over in memory for an unbounded number of digits, i.e. to perform general addition of any arbitrary pair of numbers.

## 6 CONCLUSION

We presented a learner, building on several different lines of work in the literature, that traverses a complex space of RNNs varying in both weights and architectures, in search of the network that has the minimal description length. We tested our learner on a range of sequential tasks and compared it to various RNNs from the literature. We found that our learner arrived at networks that are reliably close to the true distribution across tasks and corpus sizes. In fact, in several cases the networks achieved perfect scores. Moreover, the networks lent themselves to direct inspection and showed an explicit statement of the pattern that generated the corpus. The RNNs from the literature, on the other hand, were not just opaque but also generally performed much less reliably on the test corpora.

In current work we attempt to extend the present paper in several directions. For example, we are extending the range of generating patterns for our corpora, including dependencies from linguistic domains not considered here, such as phonotactic patterns. We are also considering training corpora that are more challenging than the ones used here in terms of size (using training corpora that are even smaller than the ones used here), noise (corrupting the training corpus using various noise patterns), and generating distribution (deviating from the simple generating distributions used here, which supported a direct comparison of simulation results with the true distribution but are overly simplistic). We are also working on extending the learner in terms of allowable units and connections. An obvious question is whether the GA search can be sufficiently efficient to support MDL, at least on relatively small corpora; we take the current results as encouraging in in this regard.

subtraction (100% accuracy), but fails to do five digits (less than 10% accuray)." Chuan Li, 'Demystifying GPT-3'. https://lambdalabs.com/blog/demystifying-gpt-3/

More formally, we estimate that the computational power needed to train an MDL learner should not exceed that of a regular RNN through backpropagation.[9]

Beyond these technical extensions, we are interested in connecting the present work more tightly with experimental results on parallel human biases and generalization preferences. MDL predicts very inclusive generalizations for small training corpora, with a narrowing as the corpus grows. Non-regularized RNNs do not make this prediction, and standard regularization schemes, which lower the values of weights but not the number of units, still predict time courses for generalization that differ from those of MDL. Another way to put it is: different learners come with different biases, biases which ought to be more visible with small training sets, and one question is how these biases relate to those of human learners. This question can be asked experimentally by looking at how human subjects generalize from very small corpora (see, e.g., Xu & Tenenbaum, 2007 for such a comparison in a slightly different setting). Beyond the comparison of MDL with alternatives that do not rely on a similar balance between simplicity and goodness of fit, we would like to explore a more detailed kind of comparison within the family of MDL models. Since the MDL score depends on the primitives that are provided and their given costs, it is possible to reason about different choices of primitives and costs in view of human generalization (see Piantadosi et al., 2016).

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

## A  NETWORK ENCODING

A network consists of $U$ units, $U$ activation functions specified for each unit, and $C$ connections (including bias connections). In order to represent a network as a binary string, the following serialization scheme is used.

### A.1  NODES

The number of nodes in the network affects its overall encoding length both explicitly and implicitly when it is used to encode other components: node numbers are used when specifying connection sources and targets, and so larger numbers require more space; and more nodes require more activation functions to be specified.

Since the number of nodes varies from network to network, their string representation cannot be of fixed length. To ensure unique readability of a network from its string representation we use a prefix-free code.

Here and throughout this section we encode integers into bit-strings using the prefix-free encoding from Li & Vitányi (2008):

$$E(n) = \underbrace{\underbrace{11111...1111}_{\text{Unary encoding of } log_2 n} \underbrace{0}_{Separator} \underbrace{1010...010}_{log_2 n}}_{n}$$

Thus for an integer $n$ its encoding length would be $2\lceil log_2 n \rceil + 1$, and the total encoding length for all units in a network would be $U(2\lceil log_2 n \rceil + 1)$.

### A.2  ACTIVATIONS

For a set of $A$ possible activation functions and $U$ units, the encoding cost for specifying all activations is $U\lceil log_2 A \rceil$. Since $A$ is constant throughout the simulation no prefix-free encoding is needed.

### A.3 WEIGHTS

To simplify the representation of weights in neural networks and to make it easier to mutate weights incrementally in the genetic algorithm, we represent each weight as a fraction made of a sign (plus/minus) and integer numerator and denominator:

$$\pm\frac{N}{D}$$

This can be serialized into bits using the following conversion and the integer encoding scheme presented above. For example, the weight $w_{ij} = +\frac{2}{5}$ would be represented as:

$$\underbrace{\underbrace{1}_{+}\underbrace{E(2) = 10...10}_{2}\underbrace{E(5) = 1110...11}_{5}}_{w_{ij}}$$

And its encoding length $|w_{ij}|$ would be the sum of:

- 1 bit for the sign.
- $2\lceil log_2 N \rceil + 1$ bits for the numerator
- $2\lceil log_2 D \rceil + 1$ bits for the denominator.

### A.4 CONNECTIONS

A connection $c_{ij}$ consists of a source unit $i$, a target unit $j$ and a weight $w_{ij}$. It can thus be encoded as:

$$\underbrace{E(i)E(j)\underbrace{0/1E(N_{ij})E(D_{ij})}_{w_{ij}}}_{c_{ij}}$$

### A.5 EXAMPLE

We'll encode the following network which consists of three units, two connections, and two weights: $\frac{1}{2} = 0.5$ and $\frac{2}{1} = 2.0$.

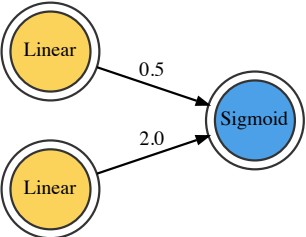

The final representation for this network is:

Note that $E(U)$ is prefixed to the string to make it possible to parse the activation part correctly.

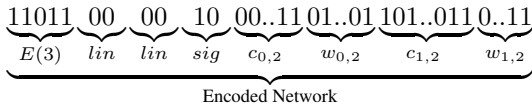

# B  GENETIC ALGORITHM

The genetic algorithm implementation for our model comprises three main components: a population representation scheme, a selection scheme and a recombination scheme. We describe here the implementation choices made for each component.

The algorithm is initialized by creating a population of $N$ random neural networks. Each network is initialized by randomizing the following parameters: number of hidden units, activation function for each unit, the set of forward and recurrent connections between the units, and the weights of each connection. In order to avoid an initial population that contains mostly degenerate (specifically, disconnected) networks, output units are forced to have at least one incoming connection from an earlier unit.

The number of hidden units and the weight numerators and denominators are randomized from a geometric prior ($p = 0.5$) to reflect their fitness based on the MDL metric.

The algorithm is run for $g$ generations, where each generation involves a selection step followed by a recombination step. During selection, networks compete for survival on to the next generation based on their fitness. A network's fitness is the inverse of its MDL score. We use Tournament Selection (Goldberg & Deb, 1991), a selection method which approximates exhaustive ranking of the population by selecting random subsets of the population of size $t$ and then selecting the best individual from each subset; this process is repeated $N$ times to select a new population with repetitions. The approximate nature of tournament selection prevents premature convergence by allowing less-than-optimal individuals to survive, and at the same time alleviates the computational load of the selection step by lowering its run time from $O(NlogN)$ to $O(N)$.

Out of the selected population, $N$ offspring networks are created through either mutation or crossover with other networks. A network is mutated using one of the following operations:

1. Add/remove a unit.
2. Add/remove a forward or recurrent connection between two units.
3. Mutate a connection by incrementing/decrementing its weight's numerator or denominator, or by flipping its sign.
4. Replace a unit's activation function with another.

These mutations make it possible to grow networks and prune them when necessary, and to potentially reach any architecture that can be expressed using our building blocks.

Two networks can also be crossed-over to create an offspring network, potentially allowing networks who perform well on different aspects of the task to share their 'genes' and create a network that performs as well as its two parents combined. We cross-over two parent networks by constructing a network which feeds the two networks in parallel and averages their outputs. While this creates a larger network which is penalized by the $|G|$ term, this has the potential to create an offspring which performs better than its parents on the $|D : G|$ term and can be pruned later. Networks are randomly selected for mutation and crossover by probabilities $p_{mutation}$ and $p_{crossover}$.

On top of the basic genetic algorithm we use the Island Model (Gordon & Whitley, 1993; Adamidis, 1994; Cantú-Paz, 1998) which divides a larger population into 'islands' of equal size $N$, each running its own genetic algorithm as described above. Once every $m_{interval}$ generations, a *migration* step occurs during which the $m_{ratio}$ top networks of each island are sent to another island in a round-robin fashion. At the receiving island, the lowest-ranking networks are replaced with the incoming migrants. The island model makes it possible to parallelize the algorithm by running each island on a different processor, while also mitigating against premature convergence, which often occurs when using large populations.

The simulation ends when all islands complete $g$ generations and the best network from all islands is taken as the solution.

All simulations reported in this paper use the following hyper-parameters:[10]

- $N = 2,000$
- $islands = 72$
- $p_{mutation} = 0.9$
- $p_{crossover} = 0.1$
- $g = 1,000$
- $t = 4$
- $m_{ratio} = 0.1$
- $m_{interval} = 20$

## C  PROOF THAT THE NETWORK FOUND FOR $a^n b^n c^n$ IS ACCURATE

The table below shows the activation values of output units and the respective probabilities of each output class (columns) after the network is fed with one of the possible sequence inputs (rows).

Given a valid $a^n b^n c^n$ sequence, it can be seen that the accuracy is 100% (except for the last of the A's, where the prediction is probabilistic) and that confidence is over 91% in all cases (for $n = 1$; for $n >= 2$ confidence rises, at 95% for $n = 2$).

| Inputs / Outputs | A | B | C | # | |
|---|---|---|---|---|---|
| # | 1/2 | 0 | 0 | 0 | (activations) |
|  | 1 | 0 | 0 | 0 | (probabilities) |
| $k^{\text{th}}$ A | 1/2 | 1/16 | -5k | 0 | (activations) |
|  | 8/9 | 1/9 | 0 | 0 | (probabilities) |
| $k^{\text{th}}$ B | sig(-16) | 1/64 | 5(k-n)+.17k | 0 | (activations) |
| $(k < n)$ | .000007 | 0.999993 | 0 | 0 | (probabilities) |
| $n^{\text{th}}$ B | sig(-16) | 1/64 | .17n | 0 | (activations) |
|  | $< 10^{-6}$ | $< 0.085$ | $> .91$ | 0 | (probabilities) |
| $k^{\text{th}}$ C | sig(-16) | 0 | .17(n-k) | 0.009 | (activations) |
| $(k < n)$ | $< 10^{-6}$ | 0 | $> .0.994$ | $< 0.0053$ | (probabilities) |
| $n^{\text{th}}$ C | sig(-16) | 0 | 0 | 0.009 | (activations) |
|  | 0.000125 | 0 | 0 | 0.999875 | (probabilities) |

## D  PROOF THAT THE NETWORK FOUND FOR ADDITION IS ACCURATE

Consider the network in Fig. 5. Call $i_n$ and $j_n$ the inputs at a time step $n$, $h_n$ the output of the hidden cell (in gray) and $o_n$ the output of the output cell.

At every time step $n$, (i) $h_n$ is the carry-over (0 or 1), with a margin for error for this carry-over of $\epsilon_{co} = .0013$ (that is, $h_n$ is in $[0, \epsilon_{co}]$ if the carry-over is 0, and in $[1 - \epsilon_{co}, 1]$ if the carry-over is 1; (ii) $o_n$ is correct with a margin of error $\epsilon = .001$, that is $o_n$ is below .001 if the $n^{\text{th}}$ digit of the sum is 0, and above .999 if it is 1.

*Proof.* Note first that the network is such that:

- $h_n = \text{sigmoid}(7x_n{}^2 - 16)$,
- $o_n = x_n{}^2 - 4\text{sigmoid}(7x_n{}^2 - 16)$,

---

[10]All simulations ran on AWS c5.18xlarge machines with 72 vCPUs each (3.0 GHz Intel Xeon).

- with $x_n = i_n + j_n + h_{n-1}$.

From there, the theorem is proven by induction. The initialization step can be easily checked (NB: $h_{-1}$ is set to 0 by convention). Suppose the result holds for $n$. Then: $x_{n+1} \in [0, \epsilon_{\text{co}}] \cup [1 - \epsilon_{\text{co}}, 1 + \epsilon_{\text{co}}] \cup [2 - \epsilon_{\text{co}}, 2 + \epsilon_{\text{co}}] \cup [3 - \epsilon_{\text{co}}, 3]$.

The fact that the result holds at the next time step $n + 1$ can be proven graphically (it can also be proven analytically, e.g., using the continuity and local monotonicity of the relevant functions):

- The left hand side of the following graph shows that $h_{n+1}$ stays within error margin $\epsilon_{\text{co}} = .00013$, and takes the appropriate value: in binary notation, the carry-over should be 0 if $x_{n+1}$ (the sum of the current inputs and the carry-over) is 0 or 1, and should be 1 if $x_{n+1}$ is 2 or 3.

- The right hand side shows that $o_{n+1}$ is correct and stays within the error margin $\epsilon = .001$: $o_{n+1}$ should be 0 if $x_{n+1}$ is 0 or 2 (a null unit digit here because 2 is 10 in binary notation), and 1 if $x_{n+1}$ is 1 or 3 (in binary notation: 1 and 11).

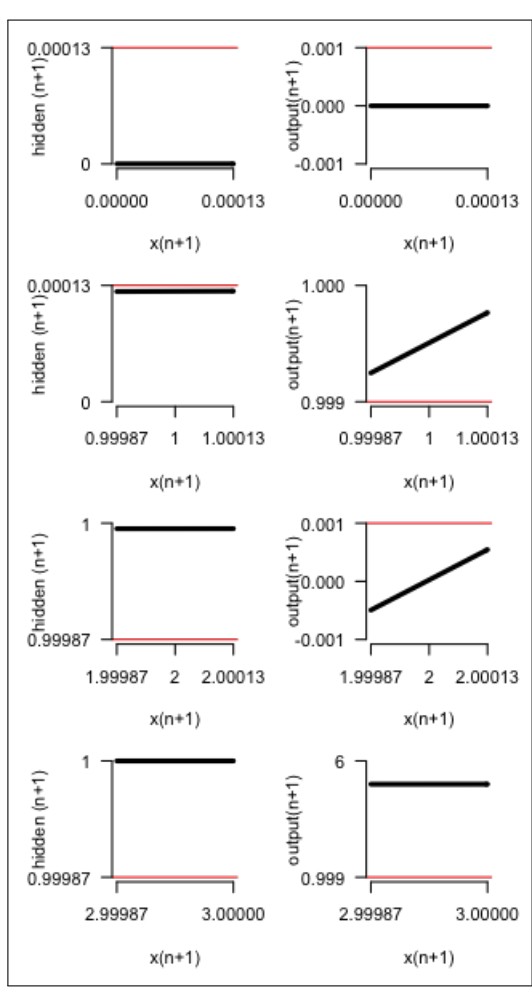

# E    CROSS-ENTROPY

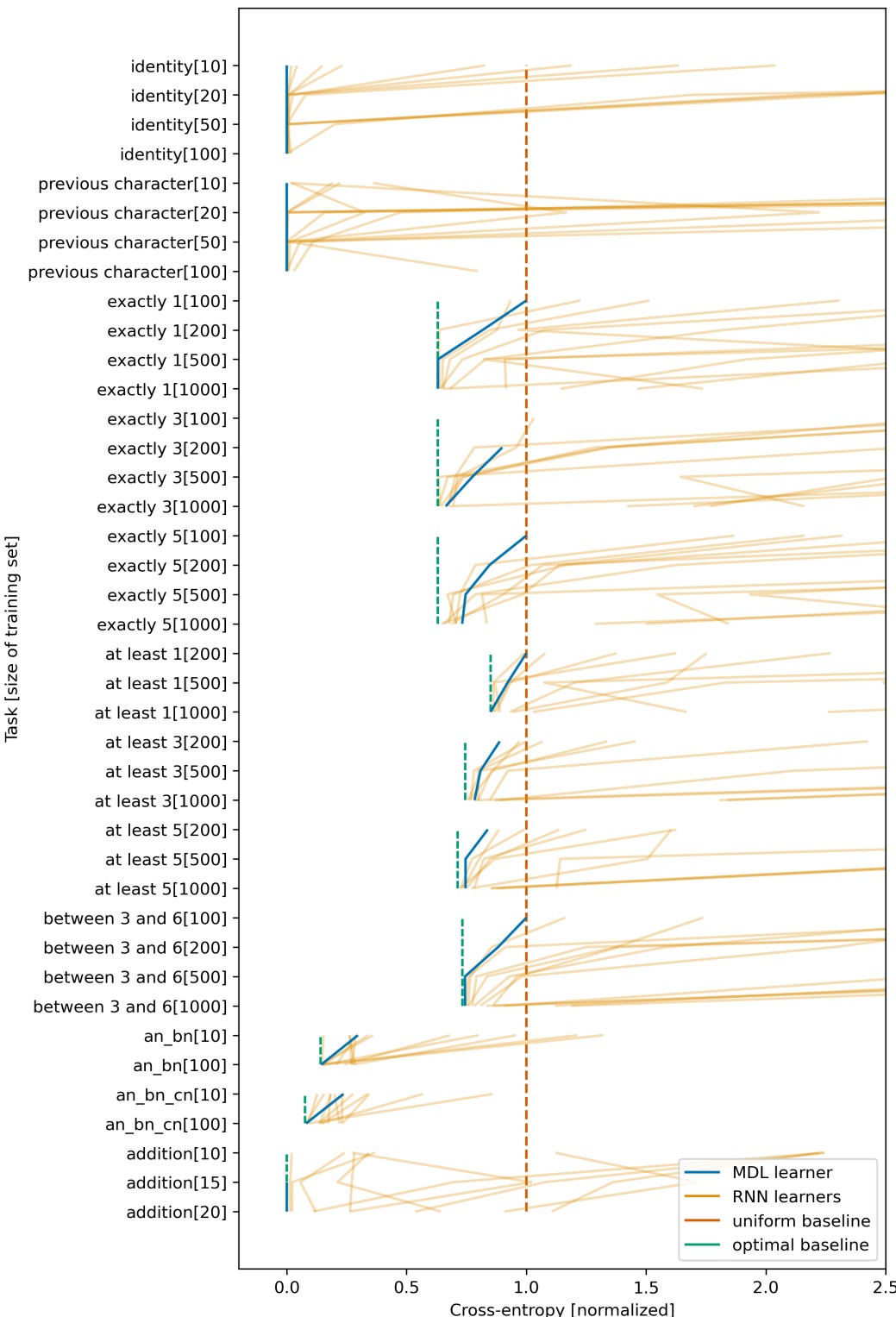

Figure 6: Cross-entropy of different learners across all the tasks presented here. For easier visualization, the cross-entropy is normalized (divided) by the cross-entropy of a learner predicting uniform probability for all outputs. At the other extreme, the green line represents an optimal baseline as the cross-entropy of a learner which would have captured the underlying task perfectly well. The gray lines represent the various RNN competitors, and the blue line is the MDL learner. The x-axis distributes the various tasks (identity, previous character, addition, exactly $n$, at least $n$, between $m$ and $n$, $a^n b^n$, $a^n b^n c^n$), for increasing sizes of the training set. Corresponding numbers are given in Appendix F.

# F    FULL RESULTS

Note on header: sub-columns labelled **2, 4, 32, 128** under *Elman cells*, *GRU cells* and *LSTM cells* are the number of network units. The second column gives the *Network Units / Seq. length* parameter for each task row.

| Corpus | Units / Seq. length | Elman 2 | Elman 4 | Elman 32 | Elman 128 | GRU 2 | GRU 4 | GRU 32 | GRU 128 | LSTM 2 | LSTM 4 | LSTM 32 | LSTM 128 | MDL learner | Optimal baseline | Uniform baseline |
|---|---|---|---|---|---|---|---|---|---|---|---|---|---|---|---|---|
| identity | 10 | 18.92 | 42.86 | 5421.90 | 9557.24 | 1189.26 | 1636.54 | 2040.69 | 828.09 | 149.06 | 231.71 | 3964.30 | 8508.80 | **0.00** | 0.00 | 1000.00 |
|  | 20 | 4.89 | 1.50 | 2221.12 | 2151.76 | 3.16 | 1.87 | 77.60 | 25.69 | 9.46 | 4.00 | 2366.09 | 1692.34 | **0.00** | 0.00 | 1000.00 |
|  | 50 | 4.30 | 1.40 | 0.50 | 0.20 | 1.13 | 1.77 | 0.40 | 0.17 | 8.48 | 3.28 | 0.18 | 199.78 | **0.00** | 0.00 | 1000.00 |
|  | 100 | 4.29 | 1.40 | 0.72 | 0.03 | 17.94 | 1.81 | 0.10 | 0.04 | 8.76 | 3.26 | 0.11 | 0.54 | **0.00** | 0.00 | 1000.00 |
| previous character | 10 | 20.80 | 223.79 | 7933.99 | 11717.89 | 13.60 | 190.95 | 7522.43 | 8150.19 | 4849.69 | 360.39 | 9696.46 | 9392.79 | **0.00** | 0.00 | 1000.00 |
|  | 20 | 324.20 | 1.71 | 0.17 | 3418.02 | 2221.69 | 0.37 | 0.81 | 5.98 | 473.52 | 1166.14 | 157.72 | 5148.17 | **0.00** | 0.00 | 1000.00 |
|  | 50 | 12.68 | 1.83 | 0.05 | 54.38 | 7.78 | 0.60 | 0.03 | 0.02 | 109.03 | 1.40 | 0.06 | 0.14 | **0.00** | 0.00 | 1000.00 |
|  | 100 | 8.51 | 1.89 | 0.05 | 0.03 | 796.60 | 1.85 | 0.03 | 0.00 | 28.94 | 1.28 | 0.06 | 0.01 | **0.00** | 0.00 | 1000.00 |
| exactly 1 | 100 | 1936.60 | 3648.23 | 7120.30 | 7033.75 | 2393.98 | 5070.70 | 6540.86 | 7406.64 | **1477.46** | 5063.69 | 7564.87 | 8918.52 | 1580.21 | 997.00 | 1580.21 |
|  | 200 | **1004.25** | 1529.65 | 8508.10 | 6537.55 | 1338.46 | 1701.06 | 6224.99 | 6930.54 | 1385.69 | 2884.95 | 5236.51 | 6022.09 | 1292.76 | 997.00 | 1580.21 |
|  | 500 | 1004.10 | 5192.52 | 1303.19 | 1439.22 | 1028.39 | 1158.60 | 4020.91 | 5767.54 | 1076.33 | 1309.92 | 4225.85 | 3034.13 | **997.24** | 997.00 | 1580.21 |
|  | 1000 | 1008.71 | 1034.17 | 2745.21 | 1446.79 | 1027.25 | 1012.29 | 4397.10 | 4653.39 | 1033.23 | 1076.22 | 2312.84 | 1805.77 | **997.25** | 997.00 | 1580.21 |
| exactly 3 | 100 | **1626.23** | 4339.55 | 5833.99 | 4689.50 | 6806.24 | 2124.06 | 7476.04 | 6501.10 | 4790.37 | 5415.79 | 6002.06 | 6309.55 | ∞ | 995.00 | 1577.04 |
|  | 200 | 1510.19 | 2108.44 | 8623.23 | 7509.59 | 4070.21 | 1152.37 | 6391.01 | 8421.04 | **1234.36** | 2023.60 | 4802.45 | 7855.51 | 1418.93 | 995.00 | 1577.04 |
|  | 500 | 1125.43 | 1059.38 | 2596.01 | 7156.36 | **1007.46** | 1016.06 | 4015.54 | 6293.62 | 1108.50 | 1165.17 | 4354.36 | 6648.57 | 1221.39 | 995.00 | 1577.04 |
|  | 1000 | 1095.00 | **999.37** | 3407.04 | 1070.40 | 999.80 | 1010.29 | 2785.53 | 6626.96 | 1062.60 | 1048.46 | 2675.78 | 2240.40 | 1046.20 | 995.00 | 1577.04 |
| exactly 5 | 100 | 2927.70 | 3388.22 | 9085.46 | 7558.98 | 3636.17 | 6239.00 | 8462.78 | 8184.39 | 4526.17 | 6223.97 | 7862.81 | 15394.40 | **1567.53** | 989.00 | 1567.53 |
|  | 200 | **1233.97** | 1598.13 | 9447.06 | 7053.78 | 1783.48 | 1675.75 | 6857.74 | 8684.41 | 1824.85 | 2543.35 | 8953.04 | 8205.14 | 1326.89 | 989.00 | 1567.53 |
|  | 500 | 1133.33 | 1089.91 | 5924.18 | 1276.61 | 1543.73 | 1417.12 | 3032.44 | 6049.26 | **1051.40** | 1233.24 | 2423.65 | 7073.05 | 1170.42 | 989.00 | 1567.53 |
|  | 1000 | 1106.63 | 1062.39 | 2348.24 | 1309.87 | **1012.54** | 1077.15 | 5109.80 | 5404.95 | 1108.79 | 1027.16 | 2891.97 | 2016.93 | 1147.83 | 989.00 | 1567.53 |
| at least 1 | 200 | **1569.11** | 2571.81 | 11890.30 | 8642.99 | 1705.38 | 3590.82 | 8630.25 | 9528.69 | 2178.12 | 2773.00 | 8013.37 | 8405.09 | 1581.79 | 1345.47 | 1581.79 |
|  | 500 | **1363.79** | 1373.41 | 2894.96 | 1701.05 | 1476.26 | 1899.24 | 6357.64 | 8643.25 | 1469.00 | 2508.96 | 3939.20 | 8420.10 | 1458.37 | 1345.47 | 1581.79 |
|  | 1000 | 1365.69 | 1408.96 | 1626.05 | 2641.63 | 1385.45 | 1472.74 | 6044.24 | 7393.21 | 1367.12 | 1491.10 | 5929.72 | 3571.62 | **1345.66** | 1345.47 | 1581.79 |
| at least 3 | 200 | **1226.46** | 1525.05 | 5818.14 | 8694.25 | 2285.18 | 2099.28 | 7430.55 | 8502.03 | 1676.90 | 3807.17 | 7399.71 | 7218.72 | 1395.39 | 1168.41 | 1569.11 |
|  | 500 | **1205.68** | 1350.51 | 7794.68 | 3319.78 | 1274.58 | 1340.37 | 6995.68 | 8227.10 | 1253.07 | 1448.67 | 1231.54 | 7912.91 | 1268.73 | 1168.41 | 1569.11 |
|  | 1000 | 1195.40 | 1194.52 | 3319.78 | 1392.78 | **1186.37** | 1253.52 | 6000.46 | 2884.16 | 1254.59 | 1330.49 | 4719.92 | 2833.47 | 1230.39 | 1168.41 | 1569.11 |
| at least 5 | 200 | 1402.24 | 1799.69 | 1360.03 | 8175.39 | 1568.88 | 1978.74 | 10265.14 | 8373.92 | 2573.32 | 2538.95 | 5307.64 | 7575.50 | **1327.42** | 1128.45 | 1581.79 |
|  | 500 | 1270.12 | 1345.81 | 8424.52 | 5141.51 | 1222.62 | 1299.93 | 5250.63 | 7394.25 | 1375.84 | 2381.27 | 1391.52 | 6469.85 | **1180.47** | 1128.45 | 1581.79 |
|  | 1000 | 1164.00 | 1185.65 | 1805.57 | 1350.10 | 1149.10 | 1237.54 | 1346.05 | 7466.91 | **1145.18** | 1218.54 | 1228.50 | 5466.44 | 1180.47 | 1128.45 | 1581.79 |
| between 3 and 6 | 100 | 1836.70 | 2746.98 | 8996.19 | 7173.70 | 5573.26 | 5079.49 | 7824.91 | 10198.21 | 9289.49 | 4838.71 | 7210.70 | 9592.36 | **1580.21** | 1159.03 | 1580.21 |
|  | 200 | **1351.46** | 2172.33 | 2190.79 | 8664.16 | 1442.93 | 3614.46 | 7121.52 | 7465.81 | 1972.58 | 2497.02 | 6667.50 | 9132.11 | 1394.29 | 1159.03 | 1580.21 |
|  | 500 | 1211.81 | 1321.87 | 1520.81 | 6611.78 | 1197.86 | 1552.33 | 7369.45 | 4211.62 | 1251.47 | 1468.81 | 5586.03 | 6234.22 | **1175.24** | 1159.03 | 1580.21 |
|  | 1000 | 1195.46 | 1198.31 | 1358.56 | 1315.84 | 1177.89 | 1276.21 | 1876.45 | 1434.92 | 1223.72 | 1235.91 | 1769.10 | 1370.39 | **1176.43** | 1159.03 | 1580.21 |
| an.bn | 10 | 8633.93 | 8727.40 | 9588.56 | 22530.83 | 9527.16 | 31675.99 | 40107.89 | 43699.78 | **5056.73** | 11907.65 | 26509.87 | 11256.88 | 9828.63 | 4680.38 | 33084.51 |
|  | 100 | 9430.55 | 8901.57 | 6872.13 | 7940.90 | 9011.99 | 9091.89 | 4832.33 | 4960.39 | 4864.96 | 4804.84 | 5047.34 | 4840.77 | **4755.00** | 4680.38 | 33084.51 |
| an.bn.cn | 10 | 21241.50 | 12297.35 | 17046.65 | 20829.44 | 14321.13 | 11392.14 | 35052.32 | 52943.52 | 11158.59 | **7866.55** | 9624.64 | 14883.77 | 14550.67 | 4680.38 | 61622.00 |
|  | 100 | 14303.12 | 13558.35 | 8158.20 | 9077.61 | 14119.33 | 8533.01 | 8786.49 | 5563.36 | 10692.51 | 5485.36 | 4873.24 | **4830.27** | 4987.38 | 4680.38 | 61622.00 |
| addition | 10 | 170772.13 | **12995.85** | 1402159.46 | 1390415.74 | 150962.48 | 176519.88 | 1403421.55 | 5335444.56 | 214553.86 | 230052.06 | 700351.53 | 5094944.62 | ∞ | 0.00 | 625000.00 |
|  | 15 | 637907.99 | 8794.11 | 673832.63 | 851465.81 | 35498.85 | 169585.51 | 435625.74 | 2112947.71 | 132402.77 | 14378.81 | 1070754.46 | 2894081.24 | **211.20** | 0.00 | 625000.00 |
|  | 20 | 164312.57 | 9050.19 | 335796.33 | 691196.19 | 75528.07 | 164882.99 | 68919.80 | 2286763.79 | 401935.11 | 14028.54 | 567813.57 | 2554335.21 | **173.07** | 0.00 | 625000.00 |

Table 2: Cross-entropy of different learners across all the tasks presented here. The winner with the lowest cross-entropy (without considering the high and low baselines in the last two columns, of course) is in bold for each task (row).

