# OpenReview forum: "Minimum Description Length Recurrent Neural Networks"
_ICLR.cc/2021/Conference — Reject_

### Official Review · AnonReviewer4 · 2020-10-21
**Interesting direction, but the experimental setup needs to be corrected**

**Rating:** 3
**Confidence:** 3

**Review:**

This paper presents an approach to training recurrent neural networks with the mix objective of minimizing cross entropy (or something similar, which is not clearly defined) and minimizing minimum description length (MDL) by using a binary representation of the network which is optimized with a genetic algorithm. The model is evaluated in contrast to RNN, GRU and LSTM  baselines on modelling a number of formal languages.

- (+) MDL is a powerful inductive bias which has been explored extensively in Machine Learning and Pattern Recognition, and it is interesting to see it revisited in the context of neural networks and in contrast to deep learning models.

- (+) The paper exhibits numerous examples in which the presented system produces simple hypothesis to model (also relatively simple) data.

- (-) Nonetheless, the studied tasks are good for a start, but insufficient to motivate the approach because actually neural networks can deal with them quite well [e.g., 1]. Moreover, the claims about the binary addition experiments are notably flawed. In particular, the task that is considered in this paper involves two bit sequences that are fed *bit by bit* in parallel to the system. This poses no serious challenge to no RNN, and in fact you can find example code directed at students that solves this task: https://github.com/mineshmathew/pyTorch_RNN_Examples. (I have personally tried this code on the same setup of using sequences of up to 20 bits and evaluating on a test set of up to 250 bits with 100% accuracy -- It only required updating a few lines to more modern versions of python/pytorch and changing the loss to BCE). The version of binary addition which is actually more challenging for RNNs is when the output is produced _after_ all operands are given [2]. Nonetheless, other neural network architectures deal with this task effectively [3].

 I can see two directions in which this work could be improved moving forward:

   - If the goal is to improve interpretability of models, the authors could aim at tackling problems in gradient-based neural network remain obscure (e.g. see https://blackboxnlp.github.io/)
   - If on the other hand the goal is improving generalization, the authors could consider tackling tasks in which neural networks have been shown to be deficient in their generalization skills [2].

- (-) Furthermore, RNN baselines might not be properly trained across the paper. First, as mentioned in the previous point, RNNs can in fact learn the binary addition task, and thus it is unclear why the authors report that they fail. Second, the authors report that the RNNs perform worse than chance on some other tasks, which could be explained by divergent training. Since no code was provided it is not possible to assess how the models were trained.

- (-) The authors enumerate some works related to theirs, but they do not comment in which ways they are similar or different from theirs.  Also, given that whole books have been written on the MDL principle, the related work section could be considerably more detailed. The relation to neural architecture search (NAS) is missing too.

- (-) It is also unclear how the training objective is quantified. It is only mentioned that |G : D| relates to cross-entropy, but no precise definition is given, nor it is detailed in which ways it differs from the former.

**Questions for the authors**

1) Do the addition RNN models reach 100% accuracy on the training data?

2) The cross-entropy numbers are quite high for binary classification problems. Could you report how you computed them?

3) For the test data you mention that you test the model on "an unseen sequence of length X". By "an unseen" you mean 1 sequence?

4) Regarding footnote 7, why not properly quantifying the number of operations needed to train each type of model?

**References**

[1] Weiss, Gail, Yoav Goldberg, and Eran Yahav. "On the practical computational power of finite precision RNNs for language recognition." arXiv preprint arXiv:1805.04908 (2018).

[2] Joulin, Armand, and Tomas Mikolov. "Inferring algorithmic patterns with stack-augmented recurrent nets." Advances in neural information processing systems. 2015.

[3] Kaiser, Łukasz, and Ilya Sutskever. "Neural gpus learn algorithms." arXiv preprint arXiv:1511.08228 (2015).

[4] Lake, Brenden, and Marco Baroni. "Generalization without systematicity: On the compositional skills of sequence-to-sequence recurrent networks." International Conference on Machine Learning. PMLR, 2018.

---

> ### Author Response · Authors · 2020-11-24
> **Response to Anonymous Reviewer 4**
>
> Thank you so much for such a detailed and constructive review.
>
> RNN competitors can indeed deal with some of the tasks here well. We agree that our claim about addition was misleading, the addition tasks that are completely beyond the scope of RNN have not been attempted here. Yet, the current results however show several interesting features, which we think are unmatched:
>
> - We did not find that a single alternative learner can deal really well with all of the tasks, that is without manual tuning for each of the tasks. In fact, beyond the lower performance of the current RNN versions we tested, we also find quite a bit of variability across these different RNN sizes/architectures. More specific RNN alternatives will be able to solve each of these tasks, but we are excited by the idea that a lot of the architecture search and training can be done in a single step, for principled reasons, and consistently across the current set of tasks.
>
> - The networks we obtain are small in absolute size.
>
> - Our networks come with a proof of accuracy (and very low cross-entropy) in general, that is above and beyond a fixed test set, but in absolute terms.
>
> Answers to specific questions:
>
> Q1: Some addition RNN models reach 100% accuracy on the training data (24/36); out of these, only 5 reach 100% on both training and test set (the latter, as mentioned, is bounded in size, compared to the general proof of correctness for the MDL net). And although the winning output is accurate, it is so with less confidence than for the MDL net (i.e., MDL leads to better cross-entropy).
>
> Q2: For both the MDL and RNN networks we calculate cross-entropy as a sum, and not as an average which is often used in training losses; this reflects the encoding length aspect of cross-entropy. To get a good sense of how to interpret those, we report the low and high baselines for each task based on this measure.
>
> Q3: Test data – sorry for the confusing phrasing: in the relevant tasks a single sequence could contain several examples of the given pattern, separated by an EOS symbol.
>
> Q4: Agreed, although what counts as a single operation depends on implementation choices. Backprop has been optimized for decades now, both at a software and hardware level, and we expect that the MDL/GA could gain much with just a little bit of research.

---

### Official Review · AnonReviewer3 · 2020-10-28
**interesting idea, preliminary finding**

**Rating:** 4
**Confidence:** 3

**Review:**


The authors proposed a new training framework for recurrent neural networks that involves updating the weights with genetic algorithm under a minimum description length principle. They evaluated it on a syntheic mini task of languange modeling and showed a better performance over classical RNNs trained by backpropagation.


Strength

+ it is a good thing to read about alternative approach to backpropagation in training deep networks.
+ tackling the interpretablity issue of deep networks with symbolic knowledge is a great approach.
+ the problem of generalization in recurrent networks is an important topic to study.
+ As in the discussion section, this line of research has great potential in future extensions if implemented and understood well.

Weakness

- the previous work offers an intriguing task on the historical attempts in applying genetic algorithms and MDL principles in neural networks. However, it would be better if what works are done in these related work. For instance, the authors wrote "These challenges were already taken up by early work on XXX", but failed to say how they tackled these challenges.
- the technical sections are very hard to follow. It is hard to decode in section 3 how the algorithm work. Although mentioned some details on the genetic algorithm in the Appendix, the main text should be self-contained.
- Following the previous point, how is the MDL metric computed exactly? Is it a bitwise estimation? If so, is the RNN here merely a boolean network?
- Why is the task descriptions in section 3.5 placed in the "Learner" section? And why are some results also included here in the method section? The writing needs some improvement to increase clarity and structure.
- Still this section, "Again, this network is transparent, the task is learned perfectly well, and no RNNs would do as well." Would the author mind explaining where this is coming from?
- The result section is also lack of important details on the tasks and evaluations. Unlike section 4.3 where the experiments are adequately introduced, the setup in section 4.1 is far too brief for the authors to grasp the task (especially if it is not usually used in the field).
- table 4.4. the cross entropy of MDL model is quite close with the best RNN in all tasks except addition (where MDL model kills) and a^nb^nc^n (where MDL model sucks). This could indicates an coincidence -- did the authors run multiple runs with different random seeds? (the plots in the appendix suggests that they only ran once).


Summary

Overall, the project introduced an interesting approach with great potential, but the results and writing appear to be preliminary. We suggest the authors to add more evaluations and improve the writing.

---

> ### Author Response · Authors · 2020-11-24
> **Response to Anonymous Reviewer 3**
>
> Thank you! Previous work: we have clarified this section showing what our current goals are compared to previous attempts. In short: (i) a systematic evaluation of vanilla MDL vs vanilla RNNs, (ii) an inspection of the non-black box delivered by the MDL learners.
>
> Technical sections: See response to Reviewer 1 about the presentation choices (the opinions are clearly coherent!). The MDL metric is explained in section 3.1 and Appendix A.
>
> “this network is transparent...” We are sorry, the phrasing was confusing, there were two points (now clarified). We meant that the way this specific MDL network works is easy to follow, and that the RNNs we tested did not perform as well.
>
> Tasks in 4.1: thank you, we have improved the presentation of these tasks.
>
> We ran a single version both for our method and RNN. We agree that this is not extensive, but note that the advantage for the MDL version is found rather consistently across tasks, and against a range of (12) competitors. There is diversity in the tasks then, and this offers some replication. Also, this replication is biased against the proposed MDL/GA learner because it has to compete with several competitors. One of the satisfying aspects of the MDL/GA approach is thus that, without manual tuning, it can compete with numerous architectures, i.e. more often than not the MDL/GA wins and if not it is really close by. On the other hand, the performance of RNN learners varies greatly from task to task.
>
> As for the a^nb^nc^n task, since the first submission of the paper we have fixed a bug in the genetic algorithm's multiprocessing synchronization step which created an instability between runs. We now report a result for this task which performs considerably better than the previous one, and is in fact proven to be 100% accurate on inputs of any length (proof provided in appendix). Note that this network is still beaten by two LSTM networks which perform slightly better in terms of cross entropy, due to a slight deviation from the perfect probability distribution in some time steps; these networks, however, don't have a proof of correctness for the general case.

---

### Official Review · AnonReviewer1 · 2020-10-28
**A study of alternative neural model training/selection with a focus on simpler and interpretalbe models. The study is interesting but scalibility is likely the main issue.**

**Rating:** 6
**Confidence:** 3

**Review:**

This paper describes a method for training neural network based on
genetic algorithms that minimizes the minimum description length
(MDL). The aim of the study is to obtain smaller, explainable networks
that learn generalizations from sequences. The paper shows that the
procedure, indeed, learns compact and interpretable networks
recognizing a number of (artificial) formal languages, as well as
addition. The networks are also compared to common RNN models
tuned/trained on the same task, indicating that the networks
discovered by MDL outperforms RNNs in majority of the experiments.


Although study uses well-known methods (MDL, genetic algorithms), the
application and premise is interesting. Without doubt, both smaller
and explainable models are good, and the paper demonstrates this
nicely with the included experiments. It is not specific to the
particular study, but strengths of the method also include learning
structure of the network as well as its weights and possibility of
learning a larger class of network architectures.

My main criticism likes within two aspects of the paper/study:

- Although the authors touch upon this in the concluding remarks, I'd
  be interested to see a more through proof or demonstration of
  scalability of the method. All the experiments provided can be
  solved by relatively simple networks. With increasing problem
  complexity, the networks size (as a result its encoding) and the
  number of alternative networks to test before finding a solution is
  expected to increase as well. In short: the usability of the method
  would be more convincing if one of the problems had a larger
  alphabet, and less well-defined solutions (e.g., a linguistic
  problem as elaborated by the authors).

- I was very surprised to find the description of the proposed
  method/model in an appendix. Although the method is relatively
  straightforward for most ICLR audience, I thin the paper needs a
  reasonably detailed description of the method in the main text. If
  the space is the issue here, one can shorten the descriptions of the
  artificial language experiments, or even push part of them to an
  appendix.

I also have some minor remarks/suggestions:

- From the description in the paper, I am not sure if the baseline
  RNNs got the same care and love as the proposed system. If not tuned
  properly, it would not be surprising that they do not necessarily
  perform well. For example, for these simple problems there is a fair
  chance that at 1000 epochs some of the networks badly overfit. Hence
  while comparing the method with RNNs the study may be basing the
  conclusions on weak baselines.

- Similar to some of the points noted above, I'd be also interested to
  see more discussion/comparison with (L1) regularized learning.

- Some readers may benefit from a better description of the network
  representation in the figures. For example, it is not clear to me
  what an output unit with no input produces (assuming that there is
  an intercept/bias term, but an explicit remark would be useful).

- A few typographic/language suggestions/issues:

    - Footnote marks should go after punctuation (e.g., footnote mark
      2).
    - The fonts on figures are sometimes unreadably small, and
      resolution is not optimal (especially fig. 6 in the appendix,
      but others may also benefit from high-resolution or vector
      graphics).
    - Aligning numbers in table 1 properly would increase their
      readability.
    - The color choice on figures (although it did not seem
      significant) was not kind to this color blind reader.

---

> ### Author Response · Authors · 2020-11-24
> **Response to Anonymous Reviewer 1**
>
> Thank you, we address the various points in order:
>
> Scalability and tasks: we agree, see responses to Reviewer 2.
>
> Presentation choices: apologies, we realize that some of the organization is non-standard. We had to put some of the technical details in the appendix. We want to emphasize here that a large part of our contribution is not technical, not about the details of the implementation of the genetic algorithm, but rather is concerned with what an out-of-the-box version of it can achieve, without manual changes, and adjustments to specific tasks.
>
> RNN alternatives: we are using a vanilla version of Genetic Algorithm search (in the words of Reviewer 2). This was done on purpose, and hopefully it means that our GA training and its RNN competitors (still the results of decades of research) were treated fairly.
>
> L1: We agree that RNN results could be improved through a variety of methods, including L1 regularization, dropout manipulations, etc. We thought it was fair to not fine-tune the systems (neither the RNN nor ours) any further though. Also, some results achieved by the MDL/GA learner cannot be achieved easily in this way, eg, proofs of accuracy had never been done and the networks would still be quite large, presumably.
>
> Cells with no input: it is absolutely correct that our networks may grow cells with no input, which is not at all standard. As we now explain (thank you), these cells behave as if their input was 0 (eg, if the activation function is a sigmoid, it will systematically output 1/2).
>
> Typos: integrated, thanks. Apologies in particular about the colors, lesson very well learned!

---

### Official Review · AnonReviewer2 · 2020-10-29
**Minimizing description by GA**

**Rating:** 4
**Confidence:** 5

**Review:**

The importance of finding minimum description RNNs is definitely unquestionable for several reasons. Thus the authors address an important problem. They decide to use GA in order to find such a representation.

My main reservation is that the paper is a vanilla application of GA without any enhancements or tailored aspects. In other words, I can't find methodological or algorithmic contributions. The experimental results also don't stand out. The tasks selected are all artificial (learning mathematical operations). To this end it would be great to include some practical cases.

---

> ### Author Response · Authors · 2020-11-24
> **Response to Anonymous Reviewer 2**
>
> Thank you! We agree with the two points made by the reviewers, and we actually tend to see them as features rather than problems.
> - It is true that the GA is completely standard, we take it as a reason to trust the robustness of the results. The outcome here is: standard methods used in a manner that makes conceptual sense, provide powerful results (without any manual fine-tuning).
> - The reviewer is also right that the tasks are currently artificial. This is useful to have a clear baseline and a clear understanding of the tasks, and therefore help us confirm that we can read through how the discovered networks are doing the task (cf. reviewer 2 comments). Another worry about this, raised elsewhere, is that the method may not scale up to bigger input/output spaces or to larger corpora. After decades of optimization, neural nets and backprop are optimized to do large tasks smoothly. About the former, it did require some computational optimization indeed, but we are now running versions e.g. MNIST tasks with no computational difficulty. About the latter, we also think that larger corpora can be handled, we take the opportunity to note though that performing well on small corpora is one of the benefits we are investigating for this approach.

---

### Decision · Program_Chairs · 2021-01-07
**Final Decision**

**Decision:**

Reject

**Comment:**

Rather than using backprop to train RNNs, this paper explores instead using GA's to train them along with an extra Minimal Description Length objective to search in the space of the simplest possible networks that can perform the task at hand. They demonstrate that the method can indeed find minimal RNNs that, when trained even on small corpus dataset of formal languages can generalize beyond the training data.

Most reviewers and myself agree that this work is really interesting, and also refreshing to see a new approach compared to the typical way of doing things. However, as we can see in the reviews, the work is not at the level of an ICLR conference submission at this point. R1, R3, and R4's reviews breaks down the points of the papers into strengths and weaknesses, and I am inclined to believe that if the authors spend more time to try to address the weakness and improve the work, this can be a great paper in the future. Although R3 gave a score of clear reject (which is too low IMO), and the authors responded to their points, I'm inclined to believe that this work warrants another revision.

Specifically, reviewers (and myself) believe that the baseline methods can indeed perform better than reported. And while, for a novel method, we don't expect the approach to scale to SOTA approaches for sequence modeling, it would improve the work vastly if there is evidence to show that it can scale to larger tasks, and give an impression that there can be a roadmap of attacking larger problems that standard methods can currently handle.

Currently, I would say the work is a great workshop paper, but would encourage the authors to continue to consider the feedback given here to work on a future revision.